# Metabolomics-Based Profiling, Antioxidant Power, and Uropathogenic Bacterial Anti-Adhesion Activity of SP4^TM^, a Formulation with a High Content of Type-A Proanthocyanidins

**DOI:** 10.3390/antiox11071234

**Published:** 2022-06-23

**Authors:** Giuseppe Mannino, Massimo E. Maffei

**Affiliations:** Department of Life Sciences and Systems Biology, University of Turin, Via Quarello 15/a, 10135 Turin, Italy; massimo.maffei@unito.it

**Keywords:** HPLC-ESI-MS/MS, MALDI-TOF-MS, radical scavenging activity, reducing antioxidant power, polyphenols, anthocyanidins, Light-Up Cell System, cellular antioxidant activity, cystitis

## Abstract

Flavonoids and proanthocyanidins (PACs) have been the subject of intense scientific investigations, both for their antioxidant properties and anti-adhesion activity against uropathogenic bacteria. We investigated the metabolomics and antioxidant capacity of SP4^TM^, a patent-pending formulation based on a mixture of plant extracts with a high content of bioactive PACs and other polyphenols. The total content of polyphenols (885.51 ± 14.19 mg/g), flavonoids (135.52 ± 8.98 mg/g), anthocyanins (54.84 ± 2.97 mg/g), and PACs (379.43 ± 12.44 mg/g) was quantified using UV-Vis assays. Use of HPLC-ESI-MS/MS revealed the presence of 5 flavanols (100.77 ± 3.90 mg g^−1^ d.wt), 11 flavonols (59.96 ± 1.83 mg g^−1^ d.wt), and 8 anthocyanins (46.96 ± 1.59 mg g^−1^ d.wt), whereas MALDI-TOF MS showed that SP4^TM^ contains PACs with one or more type-A interflavan bonds at each degree of polymerization. Regarding antioxidant properties, LUCS technology on HepG2 cells evidenced the ability of SP4^TM^ to neutralize intracellular free radicals, inhibit membrane lipid peroxidation, quench H_2_O_2_, and reduce free radicals mainly through chelating mechanism, as demonstrated by a higher FRAP value (2643.28 ± 39.86 mmol/g) compared with ABTS (139.92 ± 6.16 mmol/g) and DPPH (89.51 ± 3.91 mmol/g). Finally, the SP4^TM^ type-A PAC content strongly prevented bacterial adhesion of P-fimbriated uropathogenic *Escherichia coli* (0.23 mg/mL). In conclusion, SP4^TM^ has a strong antioxidant capacity involving multitarget mechanisms and is a potential supplement to fight urinary tract infections due to its ability to inhibit uropathogenic *E. coli* adhesion.

## 1. Introduction

During aerobic respiration, metabolic reduction of molecular oxygen to water leads to the production of reactive oxygen species (ROS), including hydroxyl radicals (^•^OH), hydrogen peroxide (H_2_O_2_) and superoxide ion (O_2_^•−^). Oxidative stress (OS) is defined as an imbalance of ROS generation in excess of the capacity of cells/tissues to detoxify or scavenge them. Oxidative stress may alter the structure/function of cellular macromolecules that eventually leads to tissue/organ dysfunction [1].

Exposure to a variety of pollutants generates ROS and OS [2] that contribute to the pathogenesis of vascular disease, cardiovascular fibrosis [1], atherosclerosis, endothelial dysfunction, platelet aggregation and heart failure [3]. Genetic predisposition and OS caused by exogenous and endogenous factors can contribute to abnormal differentiation and hyper-proliferation of keratinocytes, accordingly the development and maintenance of psoriasis [4]. Oxidative stress has also been suggested to be involved in the pathogenesis of hypertension [5] and plays an important role in diabetic endothelial dysfunction [6] and insulin resistance that affect pancreatic beta cells [7]. Oxidative stress and oxidative DNA damage are important in the pathogenesis of benign prostatic hyperplasia [8,9], while recent evidence in the field of Alzheimer disease research, OS has been identified as one of the causes of its pathogenesis [10]. Many respiratory viral infections, including COVID-19, activate innate immune responses and secretion of inflammatory cytokines that may cause death of the infected cells. An important contribution to pathogenesis of microbial infections is the development of OS [11,12,13,14].

One of the most common infectious diseases of humans is urinary tract infection (UTI), which is also a major cause of morbidity. About 50% of healthy adult women are estimated to experience at least one UTI episode in their lifetime [15]. In uropathogenic strains of *Escherichia coli*, the increased level of endogenous ROS may be responsible for adaptive mutations [16]. Antibiotic-tolerant persister cells acting on the cell wall cope with two important processes regulated by cyclic AMP in uropathogenic *E. coli*, such as OS and DNA damage. Consequently, OS response and DNA repair are relevant pathways to target in the design of persister-specific antibiotic compounds [17]. *Escherichia coli* also promotes calcium oxalate stone formation via enhancing OS and inflammation [18]. Different antimicrobial agents have beneficial role on UTI [19].

The ROS are intercepted by the antioxidative defense system, including antioxidants in food. Thus, the antioxidant defense system can be enhanced by consuming foods containing antioxidants—either at high levels or at high bioavailability [20]. Plant antioxidants play an important role in antioxidative defense systems against OS. The metabolomic approach identified in plant foods flavonoids and other dietary polyphenol antioxidants that are believed to play a major role as antioxidants and be responsible for the health benefits associated with fruit, vegetable, and red wine intake [21,22]. Examples of foods containing antioxidants include tea polyphenols [23], and grape polyphenols like resveratrol that suppress oxidative/nitrative stress [24] and mitigate the exercise induced OS [25,26]. Recent preclinical findings show that plant polyphenols provide neuroprotection against OS-induced toxicity in Parkinson’s disease [27]. The health-promoting potential of several plants is also associated with their antibacterial activity resulting from the presence of proanthocyanidins (PACs) with documented anti-adherence properties [28].

Proanthocyanidins are polymeric flavonoids known for their high free radical scavenging capacity. Proanthocyanidins show modulation of oxidative stress by a strong inhibitory effect on the murine tyrosinase and melanin synthesis [29], increase the antioxidant enzymes activity and inhibit pro-apoptotic genes expression in granulosa cells [30], rescued cognitive impairment accompanied by suppressed OS and inflammatory stress in brain regions [31], play an important role in controlling the antioxidant response of lung cancer cells via the Nrf2-ARE pathway [32], have a protective role in the abatement of doxorubicin-induced mutagenesis and cell proliferation changes [33], have promising potential to preserve retinal tissue functions via regulating oxidative stress and mitochondrial functions [34], and reverse the mRNA and protein expression of apoptosis-associated genes [35].

Type-A PACs (PAC-A) have the potential function in the prevention of prostatic cells OS damage [36], and those present in the American cranberry (*Vaccinium macrocarpon*) exert antimicrobial activities because of both anti-adhesive activity [37,38] and anti-aging effects via regulating in vivo redox state [39]. The PAC-A from *Geranium niveum* show a powerful radical scavenging in vitro activity [40] whereas PAC-A from *Euterpe oleracea* show a high antioxidant capacity, confirmed by DPPH, TEAC and ORAC methods as well as antimicrobial activities against gram-positive bacterial strains and *Candida albicans* [41]. The PAC-A from peanut (*Arachis hypogaea*) alleviates the symptoms of ulcerative colitis by regulating gut microbiota and metabolism [42], whereas PAC-A from *Aesculus turbinata* protected the retina from light exposure damage by inhibiting OS and apoptotic mechanisms [43].

The important properties of polyphenols, and particularly PAC-A, have been the target of a deep search for plant natural sources containing these compounds that eventually produced the formulation SP4^TM^, a patent pending formulation (PCT/US2022/021087) characterized by a mixture of plant extracts with a very a high content of bioactive PAC-A and other polyphenols.

The aim of this work is the metabolomic profile, the chemical and cellular antioxidant power, and the anti-adhesion activity against uropathogenic P-fimbriate *E. coli* of SP4^TM^.

## 2. Materials and Methods

### 2.1. Plant Material, Cell Lines, Reagents and Chemicals

SP4^TM^ is a patent pending formulation (PCT/US2022/021087) composed by a mixture of plant extracts containing a high content of type-A proanthocyanidins and other polyphenols. The origin of the plants is included in the relevant patent (PCT/US2022/021087).

Dulbecco’s Modified Eagle Medium (DMEM) was purchased from Thermo Fisher (Waltham, MA USA), cell lysis solution and luciferin were purchased from BPS Bioscience (San Diego, CA, USA), HepG2 human hepatocyte carcinoma, 85,011,430 cell lines, Fetal Calf Serum (FCS), ethanol (95%), Folin–Ciocolteu reagent, sodium carbonate, sodium nitrite, aluminum chloride, sodium hydroxide, potassium chloride, sodium acetate, potassium ferricyanide (99.8%), 1,1-diphenyl-2-picryl-hydrazil stable radical (DPPH^•^) (>98%), ABTS (2,2′-Azino-bis(3-ethylbenzothiazoline)-6-sulfonic acid) (98%), and phenolic compounds standards (including PAC-A2) were purchased from Sigma-Aldrich (Milan, Italy). The c-PAC standard was provided by Complete Phytochemical Solutions (Cambridge, USA). The 4-(dimethylamino)-cinnamaldehyde (DMAC), hydrochloric acid, acetic acid, acetone, acetonitrile, formic acid, potassium persulfate, 2,4,6-Tris(2-pyridyl)-s-triazine (TPTZ), and iron(III) chloride were purchased from VWR International (Radnor, PA, USA).

### 2.2. Spectophotometric Determination of Bioactive Compounds

#### 2.2.1. Content of Total Phenolic Compounds

The total phenolic content (TPC) of SP4^TM^ was determined by the Folin–Ciocalteu’s method [44], using an UV/Vis Spectrophotometer (UV-1900i, Shimadzu^®^, Milan, Italy). Briefly, 150 µL of Folin–Ciocolteu reagent was added to 300 µL of 20% (*v*/*v*) sodium carbonate and 30 µL of SP4^TM^ dissolved in 95% (*v*/*v*) ethanol using a 1:10 (*w*/*v*) ratio. Finally, the reaction mixture was made up to 1 mL with distilled water. Gallic acid (GA) was used for the preparation of the external calibration curve (y = 2.18x + 0.01; R^2^ = 0.9993; LOD: 0.0078 mg/mL; LOQ = 0.0257 mg/mL), and the results were expressed as mg GA equivalent (GAE) per gram of dry material (g^−1^ d.wt). All measurements were repeated three times.

#### 2.2.2. Content of Total Flavonoid Compounds

The total flavonoid content (TFC) of SP4^TM^ was determined by the Aluminum Chloride assay [45] using an UV/Vis Spectrophotometer (UV-1900i, Shimadzu^®^, Milan, Italy). Briefly, 30 µL of 5% (*w*/*v*) NaNO_2_ were added to 200 µL of SP4^TM^ extract dissolved as previously described. After vortexing and incubation at room temperature (RT) for 5 min, 30 µL of 10% (*w*/*v*) AlCl_3_ were added to the mixture, and samples were again incubated at room temperature (RT) for 6 min. Finally, 400 µL of 4% (*w*/*v*) NaOH were added along with 340 µL of double distilled water (ddH_2_O). After 15 min of incubation time at RT, the absorbance was read at 510 nm. Rutin was used for the preparation of the external calibration curve (y = 2.26x + 0.02; R^2^ = 0.9983; LOD: 0.0098 mg/mL; LOQ = 0.0323 mg/mL), and the results were expressed as mg Rutin equivalent (RE) per gram of dry material (g^−1^ d.wt). All measurements were repeated three times.

#### 2.2.3. Content of Total Anthocyanin Compounds

The total anthocyanin content (TAC) of SP4^TM^ was measured by the differential pH method [46], using an UV/Vis Spectrophotometer (UV-1900i, Shimadzu^®^, Milan, Italy). Briefly, 20 µL of properly diluted SP4^TM^ were added to 980 µL of 0.1 M KCl buffer (pH = 1) or to 0.04 M sodium acetate buffer (pH = 4.5). Then, the absorbances of both reaction mixtures were read at 510 nm and 700 nm. All measurements were repeated three times. The results were expressed as mg Cyanidin equivalent (CE) per gram of dry material (g^−1^ d.wt) using Equation (1).
(1)$$\mathrm{TAC}\;\left({\mathrm{mg}\;\mathrm{CE}\;g}^{-1}\;d.\mathrm{wt}\right)=\frac{\left[{\left({\mathrm{Abs}}_{510}-{\mathrm{Abs}}_{700}\right)}_{\mathrm{pH}1}-{\left({\mathrm{Abs}}_{510}-{\mathrm{Abs}}_{700}\right)}_{\mathrm{pH}4.5}\right]\times \;\mathrm{MW}\;\times 1000\times \;\mathrm{FD}\;\times {\;V}_{\mathrm{ext}}}{\varepsilon \;\times \;l\;\times \;\mathrm{PW}}$$
where: Abs is the absorbance value read at the respective wavelength (510 or 700 nm); MW is the molecular weight of cyanidin-3-glucoside (449.2 g mol^−1^); FD is the dilution coefficient used before reading the samples; ε is the molar extinction coefficient of cyanidin-3-glucoside (26,900 mM^−1^ mol^−1^); l is the path length, that in our experimental condition was 1 cm; PW is the weight SP4^TM^ used for the preparation of the extracts.

#### 2.2.4. Determination of SP4^TM^ Total PAC Content with the BL-DMAC Method

The BL-DMAC assay was performed according to the method of Prior et al. [47]. The SP4^TM^ (200–300 mg) was dissolved in 50 mL of extraction buffer. The total PACs were quantified via an external calibration curve with the use of both PAC-A2 standard and c-PAC standard, the latter has been developed for the quantification of cranberry (*Vaccinium macrocarpon*) soluble PAC [48,49]. The quantification was performed in triplicate within the linear range of calibration curves (5–30 µg/mL).

### 2.3. Characterization of SP4^TM^ Soluble Proanthocyanidin (PAC) Distribution by Matrix-Assisted Laser Desorption/Ionization-Time of Flight Mass Spectrometry (MALDI-TOF MS)

The MALDI-TOF MS analysis was performed following AOAC First Action Method: 2019.05 [50] with a Bruker UltraFlexVR III MALDI TOF/TOF mass spectrometer equipped with a SmartBeam^TM^ laser (Billerica, MA, USA). MALDI-TOF MS conditions were as previously described [50]. FlexControl, and Flex Analysis software (Bruker Daltonik GmbH, Bremen, Germany) are used for data acquisition and data processing, respectively. The percentage of A- to B-type interflavan bonds in PAC was calculated according to [50].

### 2.4. Characterization of Other SP4^TM^ Phenolic Compounds by HPLC-DAD-ESI-MS/MS

The HPLC (Agilent Technologies 1200) coupled to a diode array detector (DAD) and a 6330 Series Ion Trap Mass Spectrometry (MS) System (Agilent Technologies, Santa Clara, CA, USA) equipped with an electrospray ionization (ESI) source was used for separation and identification of compounds. The chromatographic separation was carried out as previously described [51]. Compounds were identified by comparison of the retention time (RT) and UV-VIS/MS spectra with those of authentic reference compounds or using literature data, while their quantification was performed using calibration curves of pure standard injections. For anthocyanin analysis, the binary solvent system was MilliQ H_2_O acidified with 10% (*v*/*v*) formic acid (Solvent A) and MetOH 50% (*v*/*v*) acidified with 10% (*v*/*v*) formic acid (Sigma-Aldrich, St. Louis, MO, USA) (Solvent B). The elution method involved a multistep linear solvent gradient changing from an initial concentration of 85% (*v*/*v*) A and 15% (*v*/*v*) B to 55% (*v*/*v*) A and 45% (*v*/*v*) B in 15 min. Finally, the gradient was 30% (*v*/*v*) A and 70% (*v*/*v*) B in 20 min. The initial solvent concentration was restored at the end of each run and maintained for an additional 10 min before the next injection. Sample injection volume was 10 μL. For the analysis of other polyphenolic compounds, the binary solvent system was MilliQ H_2_O acidified with 0.1% (*v*/*v*) formic acid (Solvent A) and Acetonitrile acidified with 0.1% (*v*/*v*) formic acid (Sigma-Aldrich, St. Louis, MO, USA) (Solvent B). The initial concentration of the solvents was set at 90% (*v*/*v*) A and 10% B for 5 min, then the concentration of the Solvent B was raised to 55% (*v*/*v*) A in 25 min, and finally at 70% (*v*/*v*) B in 25 min. The initial solvent concentration was restored at the end of each run and maintained for an additional 10 min before the next injection. Sample injection volume was 5 μL. Analyses were performed in triplicate.

### 2.5. Antioxidant Activity of SP4^TM^ on Human HepG2 Cells Using the Light-Up Cell System (LUCS) Method

#### 2.5.1. Neutralization of Intracellular Free Radicals by SP4^TM^

Samples of SP4^TM^ were solubilized at a final concentration of 50 mg ml^−1^ in DMEM culture medium. Solutions were then centrifuged at 8700 rpm for 10 min and experiments were performed with the supernatants. The pH of samples was compatible with the assay. HepG2 cells were seeded in 96-well plates at a density of 75,000 cells/well in DMEM medium supplemented with Fetal Calf Serum (FCS) and kept in the incubator for 24h at 37 °C in the presence of 5% CO_2_. Cells were then incubated with samples (eight concentrations obtained by serial log2 dilutions) for 4 h at 37 °C in the presence of 5% CO_2_. Experiments were done in DMEM medium without FCS. At least two independent experiments were performed each on triplicate wells. This LUCS assay measures the ability of an antioxidant to neutralize oxidative stress and the effect is measured by a delay in the kinetic evolution of fluorescence emission [52]. The antioxidant cell index (AOP index) was calculated from normalized kinetic profiles according to Equation (2):(2)$$\mathrm{AOP}\;\mathrm{index}\;\left(\%\right)=100\;–\;100\;\left(\frac{\int_{20}^{0}{\mathrm{RFU}}_{\mathrm{sample}}}{\int_{20}^{0}{\mathrm{RFU}}_{\mathrm{control}}}\;\right)$$

Dose-response curves, obtained by compiling AOP indices according to Log of the sample concentration, were submitted to a sigmoid fit according to Equation (3):(3)$$\mathrm{AOP}\;\mathrm{index}=\frac{{\mathrm{AOP}\;\mathrm{index}}_{\min}+\left({\mathrm{AOP}\;\mathrm{index}}_{\max}-{\mathrm{AOP}\;\mathrm{index}}_{\min}\right)}{1+10^{\log\left({\mathrm{EC}}_{50}-\mathrm{SC}\right)\mathrm{HS}}}$$
where SC = sample concentration, HS = Hill slope, EC_50_ (50% efficacy concentration). EC_10_ and EC_90_ were also evaluated.

#### 2.5.2. Inhibition of Plasma Membrane-Based Lipid Peroxidation by SP4^TM^

Otherwise known as CAA or DCFDA in the literature, this assay takes advantage of the presence of a diacetate (DA) group, which allows for the passage of 2′-7′-dihydrodichloro fluorescein (DCFH) diacetate (DCFHDA) across the plasma membrane. The DCFHDA is cleavable by intracellular esterases, producing the non-permeable DCFH. When cells are treated with a radical generator such as 2,2’-azobis (2-amidinopropane) dihydro-chloride (AAPH), peroxyl radicals (ROO•) are produced at the plasma membrane level, triggering transformation of intracellular non-fluorescent DCFH into fluorescent dichloro-fluorescein (DCF). Samples of SP4^TM^ were solubilized at a final concentration of 50 mg ml^−1^ in DMEM culture medium. Solutions were then centrifuged at 8700 rpm for 10 min and experiments were performed with the supernatants. The pH of samples was compatible with the assay. The HepG2 cells were seeded as above and kept in the incubator for 24 h at 37 °C in the presence of 5% CO_2_. Cells were treated with DCFH-DA at 30 μM final concentration for 1 h, washed three times, then treated with AAPH at 600 μM final concentration. Fluorescence was read on a kinetics mode every 5 min for the necessary time. Dose response standard curves are calculated according to Equation (4):(4)$$\mathrm{CAA}\;\mathrm{units}\;=100\;–\;\left(\frac{\int_{0}^{65}{\mathrm{RFU}}_{\mathrm{sample}}}{\int_{0}^{65}{\mathrm{RFU}}_{\mathrm{control}}}\;\right)\times 100$$

#### 2.5.3. H_2_O_2_ Quenching (Catalase-like) Activity of SP4^TM^

The assay is based on the intracellular presence of a DNA biosensor whose fluorescence increases in presence of H_2_O_2_. Decrease or time delayed increase of fluorescence indicates capacity of sample to neutralize (scavenge) H_2_O_2_ in a catalase-like reaction. The SP4^TM^ samples were preincubated with H_2_O_2_ at 0.34% final concentration for 1h (new SP4^TM^ sample). New SP4^TM^ sample was added with a final proportion of 14.7% to cells preincubated with TO at final concentration of 4μM for 30 min. Fluorescence was recorded on a kinetic mode for 75 min with T_0_ = TO addition time. Catalase (25 U/ mL final activity) was used as an internal positive control. Dose response standard curves are calculated according to Equation (5):(5)$$\mathrm{CAT}-\mathrm{like}\;\mathrm{index}\;\left(\%\right)=100\times \left(\frac{\int_{35}^{75}{\mathrm{RFU}}_{\mathrm{sample}}-\int_{35}^{75}{\mathrm{RFU}}_{\mathrm{control}}}{\int_{35}^{75}{\mathrm{RFU}}_{\mathrm{SAMPLE}\;\mathrm{MAX}\;\mathrm{control}}-\int_{35}^{75}{\mathrm{RFU}}_{C\;\mathrm{control}}}\;\right)$$

For all conditions, two independent experiments were performed, each on triplicate wells.

### 2.6. Chemical Antioxidant Capacity of SP4^TM^ by TEAC, FRAP and DPPH Assays

The free radical scavenging activity of SP4^TM^ was measured using both Trolox equivalent antioxidant capacity (TEAC) and DPPH assay, while the metal reducing power was evaluated via Ferric reducing antioxidant power (FRAP) assay. The TEAC and DPPH are two spectrophotometric assays monitoring the ability of antioxidants to scavenge and reduce the colored radical cations (ABTS^•+^: blue-green; DPPH^•^: violet) into a more stable and colorless compound [53,54,55]. Color loss was monitored at 725 nm and 540 nm for DPPH and ABTS assay, respectively. For each of these assays, dose-response curves were obtained by incubating the proper reaction mix (ABTS assay: 7 mM 2,2′-azino-bis(3-ethylbenzothiazoline-6-sulphonic acid + 2.45 mM K_2_S_2_O_8_; DPPH assay: 0.1mM 2,2-diphenyl-1-picrylhydrazyl) with different concentrations of the analyzed sample. For each concentration, the percentage of antioxidant activity (AA%) was then measured according to Equation (6), and the concentration determining 50% decrease of the initial radical concentration (IC_50_) was calculated accordingly.
(6)$$\mathrm{AA}\%=\frac{A_{\mathrm{blank}}-{\;A}_{\mathrm{sample}}}{A_{\mathrm{blank}}}\times 100$$
where: A_blank_ is the absorbance of blank, and A_sample_ the absorbance of sample at 517 nm.

Concerning the FRAP assay, the reaction mixture composed by 300 mM CH_3_COONa (pH = 3.6), 1 mM 2,4,6-Tris(2-pyridyl)-s-triazine (TPTZ), and 2 mM FeCl_3_ was added to properly diluted samples of SP4^TM^. After incubation at 37 °C for one hour, electrophoretic readings were taken at 595 nm against a blank not containing the sample.

For all antioxidant assays described, dose-response curves or calibration curves were created using pure Trolox as standard. Data were then expressed as IC_50_ (µg/mL) or mmol Trolox Equivalent (TE) per gram of d.wt. All analyses were performed in triplicate.

### 2.7. In Vitro Bacterial Anti-Adhesion Activity (AAA) of SP4^TM^

The SP4^TM^ was tested for in vitro bacterial anti-adhesion activity (AAA) on a per weight basis. The SP4^TM^ was suspended (60 mg/mL in PBS, neutralized with 1 N NaOH, diluted serially (2-fold), and tested for bacterial anti-adhesion activity utilizing an HRBC hemagglutination assay specific for uropathogenic P-fimbriated *Escherichia coli* according to Foo et al. [56]. In particular, tests included measuring ability of SP4^TM^ to suppress agglutination of either HRBCs (A1,Rh+) [57] or latex beads coated with synthetic P receptor analog [58] following incubation with P-fimbriated *E. coli*. The concentration at which hemagglutination activity was suppressed by 50% was recorded as an indicator of the strength of the bacterial anti-adhesion activity (AAA). Anti-adhesion assays were repeated three times and the results averaged. Controls included wells containing bacteria + PBS, HRBC + PBS, bacteria + test compound, HRBC + test compound, and bacteria + HRBC.

### 2.8. Statistical Analysis

At least three biological replicates were always performed. An ANOVA followed by Tukey–Kramer’s HSD post-hoc test (*p* < 0.05) was used to determine significant differences. A nested ANOVA was calculated for fluorometric analyses. All statistical analyses were performed by using the IBM Statistical Package for Social Sciences (SPSS) version 25 (Armonk, New York, NY, USA).

## 3. Results and Discussion

The search for plant extracts that may exert a strong cellular antioxidant activity led to the formulation of a mixture of polyphenolic compounds that are the base of the SP4^TM^ patent pending formulation (PCT/US2022/021087). Owing to the high potential of SP4^TM^, we characterized its phytochemical composition and performed a series of antioxidant assays to evaluate both its ability to neutralize free radicals and ROS at the intracellular level and its chemical antioxidant power. Surprisingly, we discovered that the SP4^TM^ formulation has a strong activity against uropathogenic P-fimbriated *E. coli* by showing a significant bacterial anti-adhesion activity.

### 3.1. SP4^TM^ Contains a High Level of Total Proanthocyanidins and Other Phenolic Compounds

The phytochemical composition of SP4^TM^ was firstly investigated through spectrophotometric assays aimed at identifying the main classes of polyphenols composing the formulation. In particular, the Folin–Ciocolteu assay, the pH differential method, and the Aluminum Chloride assay were used for the detection and quantification of the total content of polyphenols, anthocyanins, and flavonoids, respectively. In addition, the BL-DMAC assay was used for the quantification of total proanthocyanidin content.

Based on spectrophotometric data, SP4^TM^ formulation is characterized by a high content of phytochemical compounds having at least one hydroxyl group attached to an aromatic ring, as evidenced by the Folin–Ciocolteu assay (Table 1). Specifically, TPC accounts for more than 85% (w/w) of the total weight of the formulation. Comparing TPC value with that previously measured by other authors for common dried fruits [59] or dried fruit extracts [60], SP4^TM^ demonstrates a 5 to 10-fold higher content of polyphenols.

Because of the characteristic action mechanism of the Folin–Ciocolteau assay, which is based on the reduction of molybdenum and tungsten salts in blue-colored complexes, the spectrophotometric quantification is strongly affected by the presence of other compounds having at least one hydroxyl group linked to an aromatic ring, such as aromatic amino acids, proteins, and organic acids. Consequently, to selectively estimate the contribution of other phytochemicals, specific spectrophotometric assays were performed. In particular, TFC was estimated through a complexation reaction between the aluminum, the C4 keto group of the flavonoid scaffold, and the ortho-dihydroxyl groups of the A- or B-ring [61], whereas TAC was calculated by exploiting the peculiar characteristic of anthocyanidic compounds to have a red coloration in acidic (pH = 1) environment and to be uncolored in slightly acid (pH = 4.5) environment [62]. The TPAC was evaluated by reaction with the DMAC reagent that selectively reacts with condensed flavan-3-ols having the C2-C3 linkage of the flavonoid C-ring lacking in double bond [63]. The data obtained from these spectrophotometric assays suggested that although the anthocyanin and flavonoid content was modest (about 5% (*w*/*w*) and 13% (*w*/*w*) of the total dry weight, respectively), SP4TM was characterized by a very high content of total proanthocyanidins. Indeed, when the formulation was tested with the BL-DMAC method, the quantification based on the PAC-A2 standard accounted for the 38% of the total weight, whereas the same analysis performed by using c-PAC as a standard yielded a 74% value. The c-PAC standard has been developed specifically to address needs for more accurate quantification of soluble PAC in cranberry (*Vaccinium macrocarpon*) products [48]. Although c-PAC standard has not yet been validated for the quantification of proanthocyanidins from sources other than cranberry [64], the quantification based on the universally recognized PAC-A2 standard confirms that SP4TM possesses a high content of proanthocyanidins.

### 3.2. The Chemical Characterization of SP4^TM^ by MALDI-TOF MS Shows a High Percentage of Type-A- PACs and the Presence of an Oligosaccharide

The MALDI-TOF MS is ideally suited for characterizing polydispersed oligomers and is considered the mass spectral method of choice for analysis of PAC, which exhibit large structural heterogeneity. The observed masses (*m/z*) correspond to the monoisotope of the predicted compound. For example, the predicted and observed monoisotope of a PAC trimer is *(m/z* 887) representing the contribution of ^12^C, ^1^H and ^16^O to the molecule. The mass at (*m/z* 890) represents two ^13^C, or one ^13^C and one ^2^H, or one ^18^O, or two ^2^H. Mass calculating programs such as IsoPro 3.0 are used to predict the isotopic distribution of compounds and allow for comparison between predicted and observed isotopic distributions [65]. The MALDI-TOF MS separates molecular ions in the flight tube and allows baseline detection of a broad range of PAC oligomers over the degree of polymerization (DP) range of 2 to >23.

The molecular weight difference between a Type-B interflavan bond and an A-type interflavan bond is due to the loss of two hydrogen atoms (Δ 2 amu) in forming the ether linkage. The observed masses (*m/z*) correspond to the monoisotope of the predicted compound. We used the mass calculating programs IsoPro 3.0 to predict the isotopic distribution of compounds and to compare predicted and observed isotopic distributions [65].

A series of PACs, which vary only in the ratio of type-A to -B interflavan bonds, produces mass spectra with overlapping isotope patterns for each individual oligomer. An understanding of the natural abundance of C, H, and O isotopes within PAC oligomers allowed us to determine the ratios of type-A to -B interflavan bonds in PACs by applying matrix algebra to deconvolute overlapping MALDI-TOF MS isotope patterns. Precision and accuracy were validated for binary mixtures of procyanidins A2 and B2 as early reported [49].

Prior to analysis, SP4^TM^ was deionized with a strong cation exchange resin and spiked with cesium trifluoroacetate (0.01M). The PACs are detected as cesium ion adducts [M+Cs]^+^.

Positive reflectron mode MALDI-TOF MS shows the presence of a PAC DP series from the trimer (*m/z* 997) to octamer (*m/z* 2437). The labeled masses at each degree of polymerization are representative of a PAC structure that contains one or more type-A interflavan bonds (Figure 1).

Deconvolution of positive reflectron mode MALDI-TOF MS shows PACs at DP (3–4) containing predominantly one or more type-A interflavan bonds (Figure 2 and Table 2). PACs at DP (5–7) contain predominantly two or more type-A interflavan bonds, whereas PACs at DP (8) contain predominantly three or more type-A interflavan bonds.

The MALDI-TOF MS also tentatively identified an oligosaccharide series of repeating hexoses from 5 DP (*m/z* 851) to 12 DP (*m/z* 1985). The hexose has a molecular weight of 180 and the additive mass of each additional hexose is 162 atomic mass units, the difference of 18 amu is due to the loss of water during each extension of the oligosaccharide chain. Despite deionization and subsequent spiking with Cesium trifluroacetate the oligosaccharides appear to be preferably detected as sodium ion adducts [M+Na]^+^, indicating that this class of compounds has a higher affinity for sodium ion than PACs (See Appendix A). The MALDI-TOF MS results indicate that SP4^TM^ contains a PAC series in which the most predominant masses correspond to a compound with one or more type-A interflavan bonds at each degree of polymerization. The proportion of multiple type-A bonds increases as the DP increases. The large percentage content of type-A PACs, especially at high DP levels, should not be underestimated. Indeed, from a biological point of view, type-A PACs have shown a better bioactivity than type-B ones, probably due to the extra bond between subunits of type-A procyanidins, which increase their rigidity [63,66]. Moreover, considering that the ecological distribution of type-A PACs is quite limited in the plant kingdom [63], the percentage composition of SP4^TM^ formulation is very interesting. Indeed, although several and different plant genera are able to control the stereochemistry of flavanols via the reduction of leucoanthocyanidins or anthocyanins to the respective 2R,3S-flavan-3-ols or 2R,3R-flavan-3-ols thanks to the action of leucoanthocyanidin reductase (LAR; EC 1.17.1.3) or anthocyanins reductase (ANR; EC 1.3.1.77), not all plants are subsequently able to regulate the formation of type-A or type-B PACs during polymerization process [63].

Additionally, SP4^TM^ appears to contain an oligosaccharide (hexose) series from the pentamer to the dodecamer, which chemical characterization is under study and will be reported soon.

### 3.3. HPLC-DAD-ESI-MS/MS Analysis of SP4^TM^ Reveals the Presence of Other Phenolic Compounds

In order to identify and quantify other polyphenolic compounds present in the SP4^TM^ formulation, HPLC coupled with DAD and an Ion Trap Mass Spectrometer was used. The detected compounds, along with their retention times (RT), charged molecular weight (*m/z*), mass fragmentations (MS/MS), characteristic absorption wavelength (λ), and analytical quantification (mg g^−1^ d.wt) are reported in Table 3. Moreover, the corresponding chemical structure is shown in Figure 3.

Chromatographic analysis revealed the presence of 8 anthocyanins (46.96 ± 1.59 mg g^−1^ d.wt), 5 flavanols (100.77 ± 3.90 mg g^−1^ d.wt), and 11 flavonols (59.96 ± 1.83 mg g^−1^ d.wt), for a total of 204.11 ± 7.32 mg g^−1^ d.wt. Anthocyanins are a special class of phytochemicals that are characterized by a positive charge in the flavonoid scaffold. From an ecological point of view, anthocyanin compounds from flowers have the important function of enchanting pollinating insects or animals due to their wide range of coloration, which mainly depends on the pH of the environment in which they are solubilized [67].

Thanks to their UV-screening properties and antioxidant characteristics, anthocyanidic compounds are attracting the interest not only of the pharmaceutical and cosmetic industry, but also of the modern consumer, who is more and more careful to include in the diet bioactive plant compounds with potential beneficial effects on own well-being [63]. Among the identified anthocyanins, cyanidin (**#19**) and its O-methylated form peonidin (**#21**) were detected, along with their respective glycoside (**#2** and **#17**), galactoside (**#1** and **#6**), and arabinoside (**#7** and **#11**) conjugates. In accordance with the spectrophotometric analyses discussed in the previous section, LC-MS confirmed the total identified anthocyanins accounted for more than 4% (*w*/*w*) of the dried weight of the SP4^TM^ formulation. In particular, cyanidin (**#19**) and its glycosides forms [idaein (**#1**), chrysanthemin (**#2**), and cyanidin 3-O-arabinoside (**#7**)] reached almost 90% (*w*/*w*) of the identified anthocyanins. These data suggest that peonidin derivatives might contribute little to the TAC value measured via pH differential method.

In SP4^TM^ formulation, the pentahydroxyflavanone taxifolin (**#20**) and its 3-O-diglycosid (**#3**) and 7-O-diglycosid (**#4**) forms accounted for 26.26 ± 0.95 mg g^−1^ d.wt. However, the most abundant flavanol in SP4^TM^ formulation was the tetrahydroxyflavanone dihydromyricetin (**#22**) and its 3,4′-diglucoside conjugate (**#5**) (Table 3). Concerning the respective flavonols, most of the identified compounds were tetrahydroxyflavonols. In particular, six of them were sugar-conjugates of quercetin (**#18**), namely rutin (**#9**), hyperoside (**#10**), miquelianin (**#12**), isoquercitrin (**#16**), and its sambunoside derivate (**#8**). In addition, another tetrahydroxyflavonol was identified as isorhamnetin (**#24**), which is the O-methylated form of **#18**. Notably, compound **#24** was also detected in both galactosidate (**#13**) and glucosidate (**#15**) forms. The remaining two flavonols, were the tetrahydroxyflavanol myricetin (**#23**) and its rutinoside conjugate (**#14**). Most of the quantified polyphenolic compounds belong to the flavanol class, together reaching about 50% (*w*/*w*) of the total identified polyphenols. Flavanols are characterized by having both a ketone and a hydroxyl group respectively bound to C4 and C3 of the C-ring of the flavonoid core. In addition, unlike their respective flavonols, these phytochemicals have the saturated C2-C3 bond [68]. The presence of the C2-C3 double bond in other flavonoids is very crucial not for the antioxidant properties, which are mainly due to the presence of hydroxyl groups on the chemical scaffold, but for the potential bioactivity towards other biological macromolecules, such as enzymes. Flavanols, lacking this bond, are very susceptible to inactivation through the formation of strong hydrogen bonds with macromolecules. However, it has also been documented that the presence of the hydroxyl group in the C3 of the C-ring, as well as the orthogonal arrangement of the catechol, is responsible for both the modulatory effect against neutrophil oxidative burst and the self-assembly of amyloid β protein [69,70].

### 3.4. The Antioxidant Power of SP4^TM^ by Cell-Based Methods on HepG2 Human Liver Cell Models Reveals a Strong Intracellular Free Radical Quenching Activity

Having assessed the phytochemical profile of SP4^TM^, we tested its potential radical scavenging activity by using the LUCS technology on HepG2 cells. The LUCS assay measures the ability of an antioxidant to neutralize the oxidative stress at the cellular level [52]. The LUCS assay is based on the production of cellular radical species following the addition in the human hepatocellular carcinoma (HepG2) cells culture medium of thiazole orange (TO), a photo-inducible fluorescent nucleic acid biosensor [71]. The approach has been standardized on high throughput 96-well plates to allow reliable statistical analyses.

The SP4^TM^ showed an antioxidant effect in human HepG2 liver cells. Both direct and indirect antioxidant activities were assayed. Direct antioxidant activity (neutralization of intracellular free radicals) shows and antioxidant index of 984/1000 at 390.5 μg/mL whereas the CAA Index (max. 100) was 64.08 at 25 mg/mL. Comparing the CAA value obtained during our experiments with that reported by Wolfe for commonly consumed fruits [72], SP4^TM^ recorded a higher value than fruits known to have a potent antioxidant action, such as wild blueberry (292.0 ± 11.0), pomegranate (250.0 ± 10.0), blackberry (232.0 ± 11.0), strawberry (136.0 ± 18.0), and blueberry (128.0.0 ± 30.0) [73].

Figure 4 illustrates the kinetic data used to calculate the antioxidant power of SP4^TM^ in HepG2 cells (Table 4) by using the LUCS approach. The assays demonstrate a direct antioxidant activity of SP4^TM^ with a maximum effect for intracellular free radical quenching and a neutralization of intracellular free radicals with an EC_50_ of 30.11 μg/mL, followed by a H_2_O_2_ quenching effect with an EC_50_ of 596 μg/mL and to a lesser extent, an inhibition of plasma membrane based lipid peroxidation with an EC_50_ of 7.81 mg/mL.

We can range the different direct antioxidant effects as follows (max to min): intracellular free radical quenching >> H_2_O_2_ quenching >> lipid peroxidation inhibition.

A direct comparison of the obtained data with data from the literature indicates that SP4^TM^ possesses a similar antioxidant activity of BHA (31.54 μg/mL and BHT (34.25 μg/mL, twice as much the antioxidant activity of resveratrol (64.66 μg/mL, 4.5 times that of Trolox (138.50 μg/mL and 20.7 times the antioxidant activity of epicatechin [74]. The antioxidant properties observed during the experimentation assessed on the cell model system suggest a high potential of SP4^TM^ in balancing the redox status of HepG2 cells. In particular, Figure 4 shows how the formulation is able to neutralize not only free radicals (Panel A), but also to quench non-radical reactive oxygen species (Panel E), thus preventing and counteracting an excessive peroxidation of cell membrane lipids (Panel C).

### 3.5. Chemical Antioxidant Analysis of SP4^TM^ Confirm Its High Antioxidant Capacity

In order to elucidate the potential mechanism of action of the observed antioxidant effect at the cellular level, we performed three spectrophotometric assays evaluating the radical scavenging and metal reducing capacity through *in solution* assay (Table 5).

The DPPH and ABTS spectrophotometric assays are based on a decolorization principle. These molecules are powerful synthetic chemical radicals, not constitutive of a biological environment such as the cellular one. However, the various antioxidant compounds, such as polyphenols, can scavenge by electron- or hydrogen-transfer mechanisms reducing them in colorless molecules. The DPPH and ABTS assays have different specificities and kinetics. The DPPH is much more selective for small molecules, such as polyphenol aglycones, because the radical site of the molecule is stereochemically less accessible than the radical ABTS. On the other hand, ABTS has two different radical sites, and it can efficiently react with both hydrophilic and lipophilic moieties. However, it is poorly sensitive to phytochemical compounds that reduce radicals via hydrogen atom donation mechanism [75]. The mechanism of action evaluated in the FRAP assay is different. This spectrophotometric assay evaluates the ability of compounds having ortho- or meta-oriented hydroxyl groups to chelate and reduce oxidized metals. Specifically, FRAP measures the reductive capacity towards FE(III) which is reduced to Fe(II) [76]. Regarding SP4^TM^, spectrophotometric analyses revealed that FRAP >>> ABTS > DPPH (Table 5), suggesting that the main antioxidant mechanism could be reduction by chelation. Indeed, according to chromatographic analyses, almost all identified compounds had at least two ortho- or meta-oriented hydroxyl groups linked to the flavonoid A-ring, with the exception of O-methylated anthocyanin **#21** and O-methylated flavonol **#24**, along with their corresponding conjugates (**#6**, **#11**, **#13**, **#15**, and **#17**). Comparing the values obtained from the antioxidant assays on SP4^TM^ on those previously obtained from common fresh fruits [77], SP4^TM^ shows a 100 to 300-fold higher antioxidant potential. Partially, this great difference can be certainly explained by the high water content of fresh fruits, ranging between 90% and 98% of the whole fruit. However, even correcting the values for the water content of each type of fruit, SP4^TM^ would show values about 10-fold times higher.

### 3.6. SP4^TM^ Shows a Strong In Vitro Bacterial Anti-Adhesion Activity against Uropathogenic P-Fimbriated E. coli

Urinary tract infections (UTIs) are the most common bacterial diseases affecting the world population each year. Among the usual uropathogens correlated to UTIs, uropathogenic *Escherichia coli* (UPEC) is the primary cause [19]. The administration of antibiotics, including fluoroquinolones, is commonly suggested in frequent and severe infections, but the risk of these medications generally outweighs the benefits of treatment [78]. For this reason, patients with a moderate and/or recurrent clinical picture of UTI, prefer the use of natural remedies, as they perceived safer and with fewer side problems [79]. In literature, it is reported that many people use cranberry juice to prevent UTIs. Indeed, there are several therapeutic indications that suggest that this fruit has a beneficial action due to the presence of type-A PACs [56,64,80]. Consequently, in addition to evaluating the antioxidant potential of SP4^TM^ formulation, we decided to investigate its potential ability to prevent bacterial adhesion of *E. coli* by microbiological assay. Our results demonstrate that the final concentration at which anti-adhesion activity could be detected in SP4^TM^ was 0.23 mg/mL. The value of 0.23 mg/mL is consistent with results obtained from other juice-based column-extracted cranberry powders tested. A direct comparison with organic cranberry juice powder (30 mg/mL), high PAC level cranberry extracts (0.47 mg/mL), low PAC level cranberry extracts (3.5–7.5 mg/mL), bacterial anti-adhesion activity of the commercial cranberry powder (measured by MRHA activity, 0.47 mg/mL) [80,81,82] and some fruit extracts of chosen Rosaceae family species (20–80 mg/mL) [83] indicates that SP4^TM^, with 0.23 mg/mL possesses a very high antiadhesion activity towards uropathogenic P-fimbriated *E. coli*.

## 4. Conclusions

The results of this work indicate that SP4^TM^, a patented formulation based on a mixture of plant extracts with a high content of bioactive polyphenols, shows an interesting antioxidant potential when tested by both chemical and cellular assays. In particular, the formulation was able to protect HepG2 cells from OS through different mechanisms of action, such as neutralization of intracellular free radicals, inhibition of plasma membrane lipid peroxidation, quenching of H_2_O_2_, and reduction of free radicals mainly through chelating mechanism. Furthermore, phytochemical analyses, in addition to revealing the presence of 24 different polyphenolic compounds, showed that most of the identified PACs had type-A binding. Notably, the proportion of multiple type-A bonds improved as the degree of polymerization increased. The real antioxidant potential of SP4^TM^ involving multitarget mechanisms indicates that SP4^TM^ is an interesting formulation that may find applications for the nutraceutical industry, with particular reference to the production of dietary supplements aimed to alleviate OS. Due to the presence of a high content of type-A PACs, the SP4^TM^ was found to strongly exert an anti-adhesion activity against uropathogenic P-fimbriated *E. coli*, providing preliminary evidence of its potential use for the treatment of urinary tract infections.

## 5. Patents

This work is based on the formulation SP4^TM^ for which a patent is pending: PCT/US2022/021087.

## Figures and Tables

**Figure 1 antioxidants-11-01234-f001:**
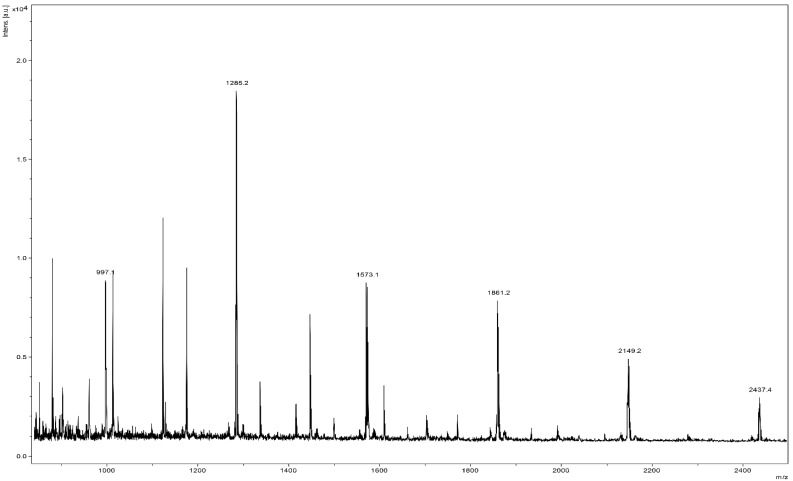
Positive reflectron mode MALDI-TOF MS of SP4^TM^ showing a PAC series from the trimer (*m/z* 997) to octamer (*m/z* 2437) in which the predominant mass at each degree of polymerization is representative of a PAC structure that contains one or more type-A interflavan bonds. Compounds are detected as Cs^+^ ion adducts [M+Cs]^+^.

**Figure 2 antioxidants-11-01234-f002:**
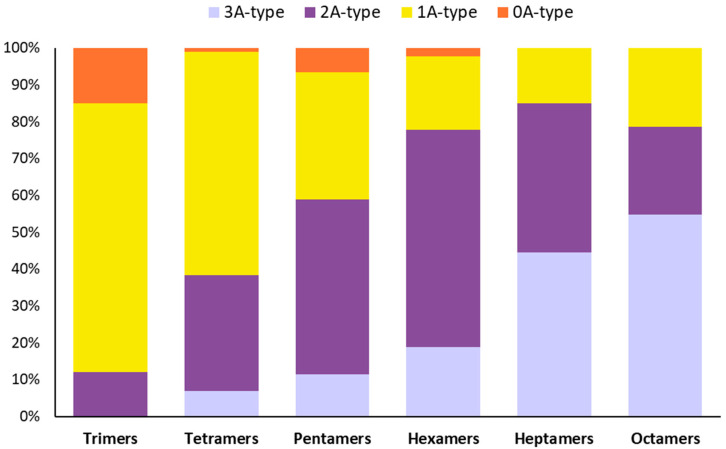
Deconvolution of positive reflectron mode MALDI-TOF MS of SP4^TM^ showing the relative type-A percentage of PACs at different degree of polymerization (DP). At 2–4 DP, PACs contain predominantly one or more type-A interflavan bonds, at 4–7 DP contain predominantly two or more type-A interflavan bonds, while at 9 DP they contain predominantly three of more type-A bonds.

**Figure 3 antioxidants-11-01234-f003:**
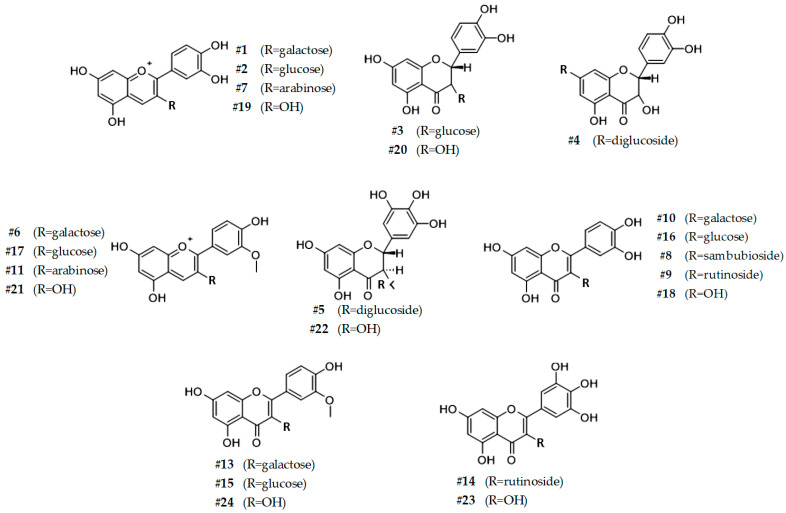
Structural formulas of the phenolic compounds characterized and quantified via HPLC-DAD-ESI-MS/MS in SP4^TM^ formulation.

**Figure 4 antioxidants-11-01234-f004:**
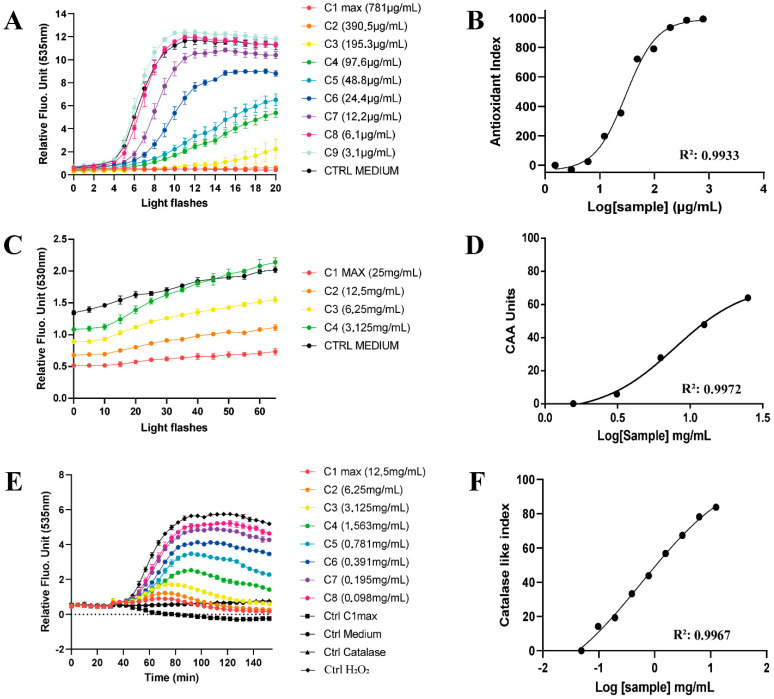
Kinetics (left panel) and dose-response (right panel) of the antioxidant power of SP4^TM^ in HepG2 cells by using the LUCS approach. (**A**,**B**), kinetics and dose-response of neutralization of intracellular free radicals; (**C**,**D**), kinetics and dose-response of plasma membrane-based lipid peroxidation; (**E**,**F**), kinetics and dose-response of H_2_O_2_ quenching (catalase-like) activity.

**Table 1 antioxidants-11-01234-t001:** Total phytochemical content of SP4^TM^ formulation based on different assays. Values are expressed as equivalent mg of reference compounds per gram of dry weight material (d.wt).

Class of Phenolics	Method Used	Content
TPC	Folin–Ciocalteu assay	885.51 ± 14.19 mg GAE g^−1^ d.wt
TFC	Aluminum Chloride assay	135.52 ± 8.98 mg RE g^−1^ d.wt
TAC	pH differential method	54.84 ± 2.97 mg CE g^−1^ d.wt
TPAC	DMAC assay	379.43 ± 12.44 mg PAC-A2 type g^−1^ d.wt
		737.54 ± 8.12 mg cPAC g^−1^ d.wt

TPC: Total Polyphenol Content; TFC: Total Flavonoid Content; TAC: Total Anthocyanin Content; TPAC: Total Proanthocyanidin Content; GAE: Gallic Acid Equivalent; RE: Rutin Equivalent; CE: Cyanidin-glucoside Equivalent; PAC-A2: Type A2 proanthocyanidin; cPAC: c-PAC proanthocyanidin standard.

**Table 2 antioxidants-11-01234-t002:** Deconvolution of positive reflectron mode MALDI-TOF MS of SP4^TM^ showing the relative percentage of type-A to -B interflavan bonds at each degree of PAC polymerization.

Degree of Polymerization	Type-A to Type-B Interflavan Bonds	Relative Percentage (%)
Trimers	2Type-A:0Type-B	12.10
	1Type-A:1Type-B	72.83
	0Type-A:2Type-B	15.07
Tetramers	3Type-A:0Type-B	6.91
	2Type-A:1Type-B	31.37
	1Type-A:2Type-B	60.70
	0Type-A:3Type-B	1.02
Pentamers	4Type-A:0Type-B	0.00
	3Type-A:1Type-B	11.38
	2Type-A:2Type-B	47.41
	1Type-A:3Type-B	34.59
	0Type-A:4Type-B	6.62
Hexamers	4Type-A: 1Type-B	0.00
	3Type-A: 2Type-B	18.88
	2Type-A: 3Type-B	58.81
	1Type-A: 4Type-B	19.98
	0Type-A: 5Type-B	2.33
Heptamers	4Type-A: 2Type-B	0.00
	3Type-A: 3Type-B	45.93
	2Type-A: 4Type-B	41.86
	1Type-A: 5Type-B	15.50
	0Type-A: 6Type-B	0.00
Octamers	4Type-A: 2Type-B	0.00
	3Type-A: 3Type-B	57.29
	2Type-A: 4Type-B	25.05
	1Type-A: 5Type-B	22.33
	0Type-A: 6Type-B	0.00

**Table 3 antioxidants-11-01234-t003:** Qualitative and quantitative phytochemical characterization of bioactive compounds in SP4^TM^ formulation. Results are expressed as mean ± SD of three different experiments carried out in triplicate, and the values are expressed as mg g^−1^ dried material (d.wt). Different lowercase letters indicate significant differences at *p* ≤ 0.05, as measured by one-way ANOVA followed by Tukey’s test.

#	RT	*m/z*	MS/MS	λ	Compound			
					Common Name	Chemical Name	CAS-ID	mg g^−1^ d.wt
**1**	22.6	449	286.9	520	Idaein	Cyanidin 3-O-galactoside	27661-36-5	0.52 ± 0.01 ^m^
**2**	24.6	449	286.9	520	Chrysanthemin	Cyanidin 3-O-glucoside	7084-24-4	4.14 ± 0.12 ^f^
**3**	24.9	465	436.7; 302.7	260, 280	Taxifolin-glucoside	Dihydroquercetin-3-O-glucoside	27297-45-6	0.12 ± 0.01 ^op^
**4**	25.0	611	574.7; 422.7	260, 280	Dihydroneohesperidin	Dihydroquercetin-7-O-diglucoside	13241-33-3	2.06 ± 0.06 ^h^
**5**	25.1	643	596.6; 490.6; 318.7	260, 280		Dihydromyricetin 3,4’-O-diglucoside		0.28 ± 0.01 ^n^
**6**	26.4	463	301	520		Peonidin 3-O-galactoside	28148-89-2	1.95 ± 0.06 ^h^
**7**	26.8	419	286.8	520		Cyanidin 3-O-arabinoside	57186-11-5	0.09 ± 0.01 ^op^
**8**	26.9	597	548.9; 301.0	260, 280		Quercetin 3-O-sambubioside	83048-35-5	0.01 ± 0.01 ^p^
**9**	27.5	609	342.7; 300.7	260, 280	Rutin	Quercetin 3-O-rutinoside	153-18-4	0.96 ± 0.02 ^l^
**10**	28.7	463	416.7; 300.7	260, 280	Hyperoside	Quercetin 3-O-galactoside	482-36-0	0.18 ± 0.01 ^no^
**11**	29.1	473	455.1, 427.0, 300.9	520		Peonidin 3-O-arabinoside	524943-91-7	0.55 ± 0.02 ^m^
**12**	29.2	477	300.7	260, 280	Miquelianin	Quercetin 3-O-glucuronide	22688-79-5	0.08 ± 0.01 ^op^
**13**	30.2	623	314.7; 299.7; 270.7	260, 280	Isorhamnetin-galactoside	3′-Methylquercetin 3-O-galactoside	6743-92-6	0.43 ± 0.01 ^m^
**14**	30.2	625	317.1	260, 280		Myricetin 3-O-rutinoside	41093-68-9	0.15 ± 0.01 ^op^
**15**	31.0	477	356.7; 314.7	260, 280	Isorhamnetin-glucoside	3′-Methylquercetin 3-O-glucoside	5041-82-7	0.15 ± 0.01 ^op^
**16**	31.9	463	416.7; 300.7	260, 280	Isoquercitrin	Quercetin 3-O-glucoside	482-35-9	1.47 ± 0.04 ^i^
**17**	32.5	463	300.9	520	Oxycoccicyanin	Peonidin 3-O-glucoside	68795-37-9	0.29 ± 0.0 1^n^
**18**	35.5	301	256.8	260, 280		Quercetin	117-39-5	11.29 ± 0.43 ^e^
**19**	36.1	287	258.9; 230.9	520		Cyanidin	13306-05-3	37.21 ± 1.29 ^b^
**20**	38.5	303	284.7	260.28	Taxifolin	Dihydroquercetin	480-18-2	24.08 ± 0.88 ^d^
**21**	48.0	301	240.1	520		Peonidin	134-01-0	2.21 ± 0.07 ^g^
**22**	48.4	318	295.4	260, 280	Ampelopsin	Dihydromyricetin	27200-12-0	74.23 ± 2.94 ^a^
**23**	51.9	317	297.9; 270.7	260, 280		Myricetin	529-44-2	30.36 ± 0.93 ^c^
**24**	52.1	315	297	260, 280	Isorhamnetin	3′-Methylquercetin	480-19-3	11.55 ± 0.35 ^e^

RT: Retention Time; *m/z*: mass-to-charge ratio; MS/MS: detected fragmentations; λ: maximum absorption wavelength; CAS-ID: Chemical Abstracts Service Identification Number.

**Table 4 antioxidants-11-01234-t004:** Antioxidant power of SP4^TM^ in HepG2 cells by using the LUCS approach. In the same column, different letters indicate significant (*p* < 0.05) differences.

Antioxidant Assay	EC_10_	EC_50_	EC_90_
Neutralization of intracellular free radicals	7.18 ^a^ μg/ mL (95% CI: 3.36–12.13 μg/mL)	30.11 ^a^ μg/ mL (95% CI: 23.23–39.08 μg/mL)	126.30 ^a^ μg/ mL (95% CI: 74.62–267.7 μg/mL)
Inhibition of plasma membrane-based lipid peroxidation	2.26 ^b^ mg/ mL (95% CI: 0.021–238.3)	7.81 ^b^ mg/ mL (95% CI: 1.105–55.2)	27.02 ^b^ mg/ mL (95% CI: 0.058–12548)
H_2_O_2_ quenching (catalase-like) activity	7.17 ^a^ μg/ mL (95% CI: ND)	596.00 ^c^ μg/ mL (95% CI: ND)	49.53 ^c^ mg/ mL (95% CI: ND)

**Table 5 antioxidants-11-01234-t005:** Chemical antioxidant activity of SP4^TM^. Values are expressed as µmol of Trolox Equivalent (TE) per gram of dried material and as IC_50_ (in mg ml^−1^).

Assay	µmol TE/g	IC_50_
ABTS	139.92 ± 6.16	2.85 ± 0.03
DPPH	89.51 ± 3.91	4.46 ± 0.12
FRAP	2643.28 ± 39.86	

ABTS: 2,2’-azino-bis(3-ethylbenzothiazoline-6-sulfonic acid; DPPH: 2,2-diphenyl-1-picrylhydrazyl; FRAP: Ferric Reducing Antioxidant Power.

## Data Availability

The data presented in this study are available on request from the corresponding author, due to patent pending (PCT/US2022/021087).

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
