# Peer review of "Metabolomics-Based Profiling, Antioxidant Power, and Uropathogenic Bacterial Anti-Adhesion Activity of SP4TM, a Formulation with a High Content of Type-A Proanthocyanidins"

_antioxidants, 2022, doi:10.3390/antiox11071234_

Round 1

Reviewer 1 Report

The paper submitted by Mannino et al. is focused on the analysis of procyanidins in products SP4 and the evaluation of its bioactivities in vitro.

The paper is well-written and all experiments are correctly planned and performed. In my opinion, the manuscript should be published in Antioxidants after addressing some minor comments.

1) table 2 - please include statistical analysis - please add SD values

2) In table 4 please indicate whether there are statistically significant differences between obtained values using the statistic test

3) please indicate in table 3 how exactly compounds were fully identified - did you use any chemical standards? if so please mark it in table 3.

Author Response

The paper submitted by Mannino et al. is focused on the analysis of procyanidins in products SP4 and the evaluation of its bioactivities in vitro. The paper is well-written and all experiments are correctly planned and performed. In my opinion, the manuscript should be published in Antioxidants after addressing some minor comments.

We thank the reviewer for his/her evaluable comment. Below, we reply to the comments in a point-by-point manner.

table 2 - please include statistical analysis - please add SD values

Table 2 cannot include standard deviations because, although the experiment was performed in triplicate, the values here reported are  expressed as percentages. Moreover the aim was to indicate the percentage of distribution of the different classes of polymers with no intention to make a statistical comparison.

In table 4 please indicate whether there are statistically significant differences between obtained values using the statistic test

We thank the reviewer again for the comment. Following his/her directions, we added the statistical significances between the different data.

please indicate in table 3 how exactly compounds were fully identified - did you use any chemical standards? if so please mark it in table 3.

The information requested by the reviewer is already contained in the relevant materials and methods section of the main text. We believe that repeating this information in the table caption would make it redundant and extremely long. We sincerely hope the reviewer can understand our reasons.

Reviewer 2 Report

Manuscript is dealing with metabolomic profile, the chemical and cellular antioxidant 100 power and the anti-adhesion activity against uro-pathogenic P-fimbriate E. coli of SP4TM. Manuscript is interesting, professionally written and all statements are supported by relevant data. However, minor revisions are necessary before acceptation.

Please include more informations about SP4. What plants? Not need to have proportions!

Please conclude according to the main objectives of the research and inclupe more practical applications of your findings.

Author Response

Manuscript is dealing with metabolomic profile, the chemical and cellular antioxidant 100 power and the anti-adhesion activity against uro-pathogenic P-fimbriate E. coli of SP4TM. Manuscript is interesting, professionally written and all statements are supported by relevant data. However, minor revisions are necessary before acceptation.

We thank the reviewer for his/her evaluable comment. Below, we reply to the comments in a point-by-point manner.

Please include more informations about SP4. What plants? Not need to have proportions!

We apologize to the reviewer, but this request cannot be satisfied. Currently, an application to register the formulation as patent is pending. We had to sign a non disclosure agreement to keep the formulation reserved until the patent will be public. We sincerely hope that the reviewer can understand our reasons.

Please conclude according to the main objectives of the research and include more practical applications of your findings.

We thank the reviewer for the suggestion. Following this suggestion, we have added sentences in the conclusion section to better explain the potential industrial applications of SP4TM.